# Giant room-temperature nonlinearities in a monolayer Janus topological semiconductor

Jiaojian Shi [1,2,13], Haowei Xu [3,13], Christian Heide [4,5], Changan HuangFu[6], Chenyi Xia[1,2], Felipe de Quesada [1,2], Hongzhi Shen[7], Tianyi Zhang [8], Leo Yu[9], Amalya Johnson [1], Fang Liu [5,10], Enzheng Shi[7], Liying Jiao [6], Tony Heinz[5,9], Shambhu Ghimire [5], Ju Li [3,11], Jing Kong [8], Yunfan Guo[12] ✉ & Aaron M. Lindenberg [1,2,5] ✉

Nonlinear optical materials possess wide applications, ranging from terahertz and mid-infrared detection to energy harvesting. Recently, the correlations between nonlinear optical responses and certain topological properties, such as the Berry curvature and the quantum metric tensor, have attracted considerable interest. Here, we report giant room-temperature nonlinearities in non-centrosymmetric two-dimensional topological materials—the Janus transition metal dichalcogenides in the $1T'$ phase, synthesized by an advanced atomic-layer substitution method. High harmonic generation, terahertz emission spectroscopy, and second harmonic generation measurements consistently show orders-of-the-magnitude enhancement in terahertz-frequency nonlinearities in $1T'$ MoSSe (e.g., > 50 times higher than $2H$ MoS$_2$ for 18[th] order harmonic generation; > 20 times higher than $2H$ MoS$_2$ for terahertz emission). We link this giant nonlinear optical response to topological band mixing and strong inversion symmetry breaking due to the Janus structure. Our work defines general protocols for designing materials with large nonlinearities and heralds the applications of topological materials in optoelectronics down to the monolayer limit.

Advances in nonlinear optics empower a plethora of applications, such as attosecond light sources based on high harmonic generation (HHG) and photodetectors for sensitive terahertz (THz) detection at elevated temperatures[1–4]. Inherently, the nonlinear optical properties of materials are connected with their magnetic structures[5,6], crystalline symmetries[7,8], and electronic band topologies. In particular, nontrivial band topologies lead to exotic electronic dynamics and enhanced optical responses[9–14]. Notable examples include anomalous HHG in various classes of topological materials[15–18]. The observation of enhanced optical responses in topological materials

[1]Department of Materials Science and Engineering, Stanford University, Stanford, CA 94305, USA. [2]Stanford Institute for Materials and Energy Sciences, SLAC National Accelerator Laboratory, Menlo Park, CA 94025, USA. [3]Department of Nuclear Science and Engineering, Massachusetts Institute of Technology, Cambridge, MA 02139, USA. [4]Department of Applied Physics, Stanford University, Stanford, CA 94305, USA. [5]Stanford PULSE Institute, SLAC National Accelerator Laboratory, Menlo Park, CA 94025, USA. [6]Key Laboratory of Organic Optoelectronics and Molecular Engineering of the Ministry of Education, Department of Chemistry, Tsinghua University, 100084 Beijing, China. [7]School of Engineering, Westlake University, 310024 Hangzhou, China. [8]Department of Electrical Engineering and Computer Science, Massachusetts Institute of Technology, Cambridge, MA 02139, USA. [9]E. L. Ginzton Laboratory, Stanford University, Stanford, CA 94305, USA. [10]Department of Chemistry, Stanford University, Stanford, CA 94305, USA. [11]Department of Materials Science and Engineering, Massachusetts Institute of Technology, Cambridge, MA 02139, USA. [12]Key Laboratory of Excited-State Materials of Zhejiang Province, Department of Chemistry, State Key Laboratory of Silicon and Advanced Semiconductor Materials, Zhejiang University, 310058 Hangzhou, China. [13]These authors contributed equally: Jiaojian Shi, Haowei Xu. ✉e-mail: yunfanguo@zju.edu.cn; aaronl@stanford.edu

have been found primarily in three-dimensional systems until now[12,13,15–17,19]. Designing two-dimensional (2D) platforms with strong optical responses is advantageous for optoelectronic applications at the nanoscale with easy controllability and scalability, but so far is limited to topologically trivial materials such as graphene[20] and 2H-phase transition metal dichalcogenides (TMDs)[21]. A promising topologically nontrivial candidate are the monolayer Janus TMDs (JTMDs) in the distorted octahedral (1T') phase[3]. Similar to 1T' pristine TMDs[22–24], 1T' JTMDs are topologically nontrivial with an inverted bandgap in the THz regime (tens of meV). Generally, a topologically protected band structure and small electronic bandgap result in larger Berry connections, larger electronic interband transition rate, and thus stronger optical response. In addition, by replacing the top layer chalcogen atoms (e.g., sulfur) in the monolayer 1T' TMDs with a different type of chalcogen (e.g., selenium), the resulting Janus structure has strong inversion asymmetry and electric polarization[25,26], which can further improve the nonlinear optical response.

In this work, we report experimental observations of giant nonlinearities at THz frequencies in monolayer 1T' JTMDs, which are synthesized via a room-temperature atomic-layer substitution (RT-ALS) method[27] under ambient conditions. It is revealed that, although the electromagnetic interaction occurs only in a single monolayer flake of 1T' MoSSe (~10–20 μm in transverse size), the generation of mid-infrared high harmonics, THz emission, and infrared second harmonic generation are all exceptionally efficient. Further comparison with topologically trivial TMDs and theoretical analyses indicate that the keys to such giant THz-frequency nonlinearities are strong inversion symmetry breaking and topological band mixing. Our results suggest that 1T' JTMDs is a promising material class that could lead to an era in THz/infrared sensing using atomically-thin materials. Our results also deepen the understanding of the fundamental mechanisms underlying strong nonlinear optical responses, which could have a profound influence in, for example, room-temperature THz detection and clean energy harvesting via the bulk photovoltaic effect[2,28].

## Results

### Multimodal nonlinearity characterization of 1T' MoSSe

The schematic illustration of multimodal characterization methods is shown in Fig. 1a. Our experiments investigated the THz-frequency nonlinearities of monolayer 1T' MoSSe with three different techniques, i.e., high harmonic generation (HHG)[29,30], THz emission spectroscopy (TES), and second harmonic generation (SHG). These techniques access nonlinear coefficients with different orders (2nd to 18th order) and spectral ranges (THz to infrared). As a comparison, we also studied the responses of monolayer 2H MoSSe, 1T' MoS₂, and 2H MoS₂ under the same measurement conditions. Such combined information unequivocally indicates giant THz-frequency nonlinearities for 1T' MoSSe. As shown in Fig. 1b, c, 1T' MoSSe and MoS₂ have distorted octahedral structures, with band inversion between metal $d$-orbitals and chalcogen $p$-orbitals[22]. In contrast, the 2H phase is characterized by a trigonal prismatic structure and is topologically trivial. In this work, Janus 1T' MoSSe and 2H MoSSe (Fig. 1d) are respectively converted from 1T' MoS₂ and 2H MoS₂ by the room-temperature atomic-layer substitution method[2]. Highly reactive hydrogen radicals produced by a remote plasma were used to strip the top-layer sulfur atoms. Meanwhile, selenium vapor was supplied in the same low-pressure system to replace the missing sulfur, resulting in the asymmetric Janus MoSSe in 1T' phase and 2H phase. To confirm the fidelity of material conversion, Raman scattering measurements were performed due to their sensitivity to the crystal lattice structure (Fig. 1e). For Janus 2H MoSSe, the positions of the $A_{1g}$ mode (~288 cm⁻¹) and $E_{2g}$ mode (~355 cm⁻¹) are consistent with literature results[2]; Meanwhile, the multiple $A'$ modes of Janus 1T' MoSSe located at ~226.2 cm⁻¹, ~298.4 cm⁻¹, ~429.8 cm⁻¹ agree well with the theoretical calculations as well, indicating the successful material substitution.

### Efficient high-harmonic generation

We first show highly efficient HHG from a single monolayer flake of 1T' MoSSe. The excitation source for HHG is mid-infrared (MIR) pulses with in-plane linear polarization at 5-μm wavelength, 1-kHz repetition rate, and ~20 MV/cm peak field strength (setup schematic shown in Supplementary Fig. 1). The HHG image acquired in 1T' MoSSe (Fig. 2a)

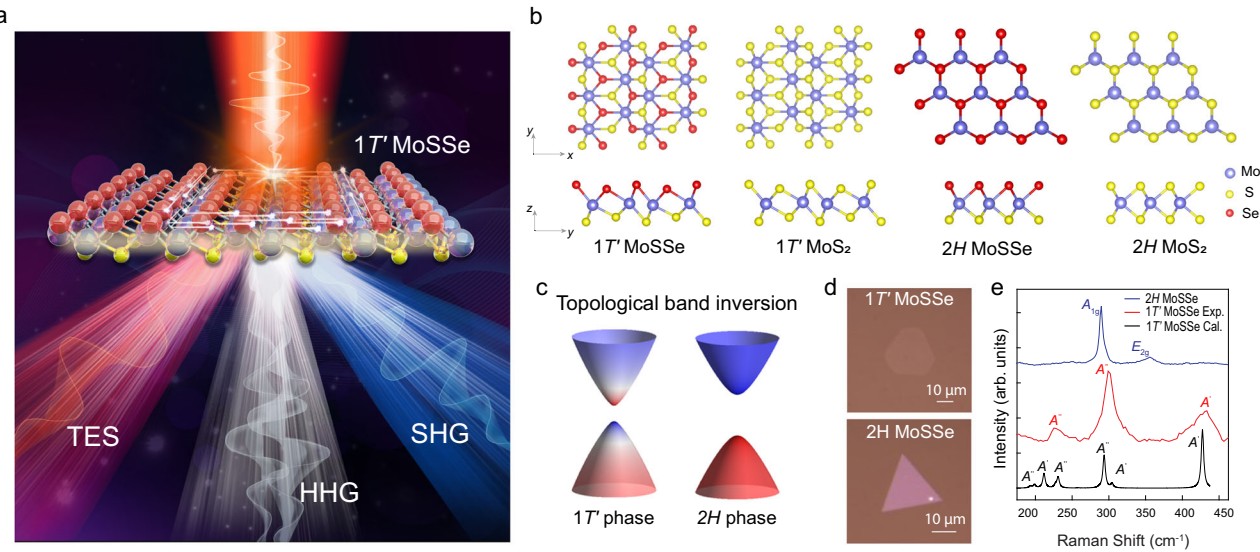

**Fig. 1 | Giant THz-frequency nonlinearities in 1T' MoSSe. a** Schematic illustration of THz emission spectroscopy (TES), mid-infrared high-harmonic generation (HHG), and near-infrared second harmonic generation (SHG) in 1T' MoSSe. **b** The lattice structure of 1T' MoSSe, 1T' MoS₂, 2H MoSSe, 2H MoS₂. **c** Schematic illustration of the topological band inversion in the 1T' phase (left) compared with the topologically trivial band structure of the 2H phase. The colormap (red and blue)

indicates the wavefunction contributions from contributed by different atomic electron orbitals (e.g., chalcogen $p$ and metal $d$ orbitals.), and the 1T' phase exhibits a hybridization between the original valence and the conduction bands. **d** Optical images of 1T' MoSSe (top) and 2H MoSSe (bottom). **e** Experimental and theoretical Raman spectrum of 1T' MoSSe and experimental Raman spectrum of 2H MoSSe.

contains at least up to 18th order response, limited by our detection scheme. The even-order HHG, which is absent in bulk TMDs[30], is a direct consequence of the broken spatial symmetry of the monolayer Janus systems. We varied the incident MIR polarization and observed nearly perfect cancellation of HHG intensity at specific angles to one of the crystallographic axes, indicating the HHG signal originates from a single flake instead of an average over many flakes with random orientations (shown in Supplementary Fig. 4). This is consistent with the laser spot size ($1/e^2$ size) ~100 μm and the sparse flake-flake spacing (shown in Supplementary Fig. 4). The HHG intensity of single flake $1T'$ MoSSe are further compared with that of millimeter-scale $2H$ MoS₂ under the same condition. Despite the irradiated flake being generally ~10 times smaller than the laser spot, the HHG of $1T'$ MoSSe is over an order of magnitude stronger than that of the millimeter-scale $2H$ MoS₂ with 100% coverage (Fig. 2b–d)[31]. The strong THz nonlinearity of $1T'$ MoSSe is further confirmed by comparing it with other reference samples ($2H$ MoSSe and $1T'$ MoS₂). Figure 2e

shows the HHG spectrum of $2H$ MoSSe, which has much weaker even-order harmonics than those of $1T'$ MoSSe. Meanwhile, the HHG of $1T'$ MoS₂, which is also topological nontrivial[22], shows relatively strong odd-order harmonics but no even-order harmonics, due to the inversion symmetry (Fig. 2f). Further semi-quantitative HHG efficiency comparison with other literature[16,30] is summarized in Table 1 showing clear advantages of $1T'$ MoSSe over most solid-state bulk or film samples.

**Enhanced terahertz emission and second-harmonic generation**
Dramatic enhancements in the TES and SHG measurements further validate giant nonlinearities in $1T'$ MoSSe. Figure 3a shows the TES measurements under 800-nm laser excitation (details in Supplementary Fig. 2) on four kinds of chemical vapor deposition (CVD) grown samples ($1T'$ MoSSe, $1T'$ MoS₂, $2H$ MoSSe, and $2H$ MoS₂), among which $1T'$ MoSSe shows distinctly higher THz emission efficiency. We do not observe a detectable signal in $1T'$ MoS₂ with the same excitation

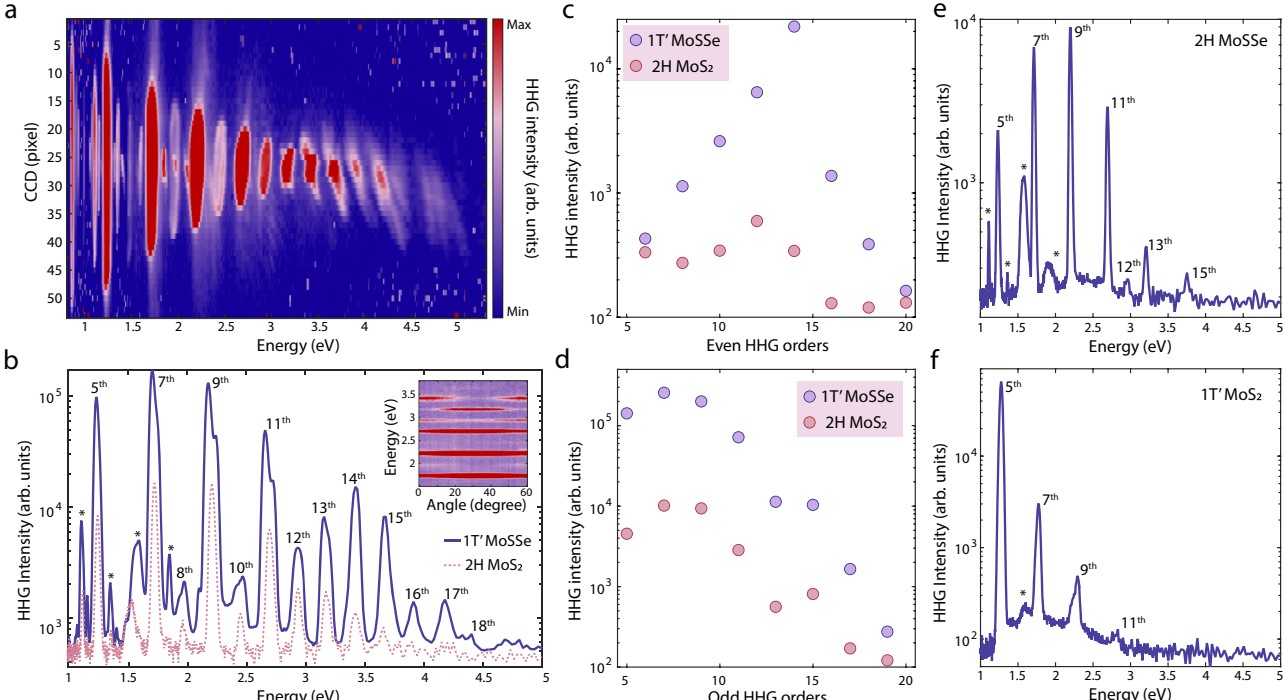

**Fig. 2 | Efficient mid-infrared high harmonic generation (HHG) in $1T'$ MoSSe.**
**a** HHG images of $1T'$ MoSSe observed by CCD camera. HHG extends up to ~5 eV and is limited mainly by the cutoff of detection optics (e.g., aluminum mirrors and grating). **b** HHG spectrum of $1T'$ MoSSe shown as the blue curve is over an order of magnitude stronger than the HHG from macroscopic monolayer $2H$ MoS₂ shown as the red dashed line. The inset shows $1T'$ MoSSe HHG intensity as a function of MIR incident polarization angles. The cancellation of a few orders at some polarization angles indicates the signal is generated from a single flake. **c, d** Quantitative comparison of HHG intensity between $1T'$ MoSSe and $2H$ MoS₂. Although the coverage

of the $1T'$ MoSSe sample is ~10 times lower than wafer-scale $2H$ MoS₂, the HHG signal is enhanced by over an order of magnitude, and orders as high as 18 or more are observed in $1T'$ MoSSe while only up to 15th order can be observed in macroscopic monolayer $2H$ MoS₂. **e** The HHG spectrum of $2H$ MoSSe taken under the same conditions. The asterisks are artifacts due to high-order spectrometer diffraction. **f** HHG spectrum of $1T'$ MoS₂. Even orders of $2H$ MoSSe and $1T'$ MoS₂ are weak and indetectable compared with $1T'$ MoSSe. The asterisks are artifacts due to high-order spectrometer diffraction.

**Table 1 | High-harmonic generation efficiency comparison of different materials**

| Material | Phase | Thickness[(i)] (nm) | Pump field[(ii)] (V/Å) | Excitation λ (nm) | Bandgap (eV) | 17th order rate (unit)[(iii)] | 18th order rate (unit)[(iv)] | Reference |
|---|---|---|---|---|---|---|---|---|
| $1T'$ MoSSe | Solid | ~1 | 0.2 | 5000 | ~0.01 | ~100 (1) | ~20 (0.2) | This work |
| Bi₂Se₃ | Solid | ~1–10 | 0.25 | 5000 | ~0.3 | <0.01 (10⁻⁴) | <0.002 (2*10⁻⁵) | Ref. 16 |
| MoS₂ | Solid | ~1 | 0.4 | 5000 | ~1.8 | <1 (10⁻²) | <0.1 (10⁻³) | Ref. 30 |

Note: (i) Effective thickness is estimated and adopted here. (ii) The selected field strength of the excitation field. (iii)–(iv) The HHG efficiency is estimated and presented by emitted photon numbers/effective thickness/pump field. The numbers in the () are normalized to the 17th order for $1T'$ MoSSe. Note the linear normalization to the pump field underestimates the difference between $1T'$ MoSSe and other samples since HHG is a nonlinear process and incident field strength in $1T'$ MoSSe is relatively low.

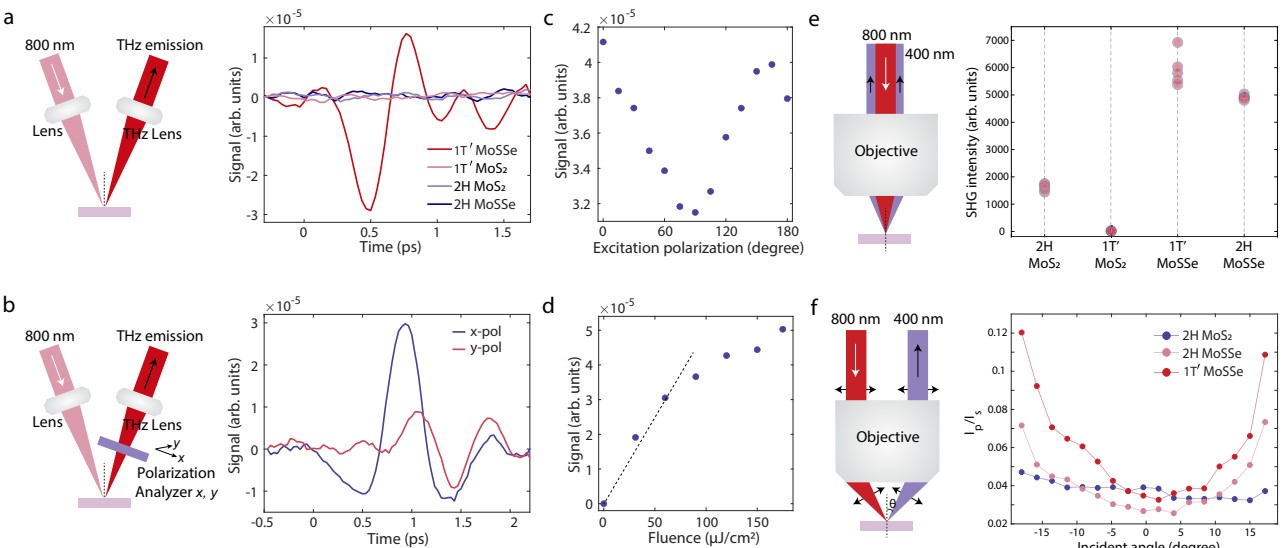

**Fig. 3 | THz emission spectroscopy and second harmonic generation of 1 $T'$ MoSSe, 1 $T'$ MoS₂, 2H MoSSe and 2H MoS₂. a** Left plot shows a schematic of THz emission setup. 1 $T'$ MoSSe shows a dramatically enhanced THz emission signal compared with three other types of monolayer TMDs (right plot). **b** Left plot shows a schematic of polarization analysis of THz emission, which shows the emission contains a major in-plane component with a possible out-of-plane contribution (right plot). **c** Dependence of peak THz field on excitation polarization. No polarizers were used for the emitted THz signal. **d** Scaling of THz emission shows a linear dependence to incident power at low fluences and saturation at higher fluences. **e** Left plot shows a schematic of second harmonic generation with normally incident 800-nm excitation. The white arrow marks the incident 800-nm beam and the black arrow marks the SHG light. The right plot shows SHG intensity of five

different flakes in each sample and shows SHG is enhanced in 1 $T'$ MoSSe and 2H MoSSe compared with 1 $T'$ MoS₂ and 2H MoS₂. **f** The left plot shows the schematic of the angle-resolved SHG setup that measures out-of-plane dipole. The white arrow marks the incident 800-nm beam and the black arrow marks the SHG light. The beam position at the objective back aperture is scanned perpendicular to the incident beam direction with a motorized stage, which tunes the incident angle $\theta$ accordingly. The right plot shows the angle-dependent SHG intensity ratio between $p$ and $s$ polarization ($I_p$ and $I_s$) in 1 $T'$ MoSSe, 2H MoSSe, and 2H MoS₂. In the 1 $T'$ MoSSe and 2H MoSSe, the $I_p/I_s$ ratio increases at non-normal incidence angles, indicative of out-of-plane dipoles. In 2H MoS₂, almost no change is observed as the incident angle varies.

fluence, consistent with its centrosymmetric structure, which forbids second-order nonlinear response. The weak TES signal in 2H MoS₂ has been attributed to an inefficient surface photocurrent[32,33]. The augmented TES in 1 $T'$ MoSSe aligns with the theory that 1 $T'$ TMDs exhibit giant nonlinearities at THz frequencies[3]. The polarization analysis of the THz emission (Fig. 3b) reveals the emitted radiation is mainly polarized in the lab-frame $x$-direction and contains a slightly weaker $y$-direction component (axis definition shown in Fig. 3b). Based on our experimental configuration, the $x$-direction emission has contributions from both in-plane and out-of-plane photoresponses, while the $y$-direction emission originates only from in-plane photoresponses. Thus, the observation of emission in the $y$-direction indicates the existence of an in-plane current contributing to the TES signal, but the stronger emission in the $x$-direction indicates there are likely significant contributions as well from out-of-plane currents. Further experiments are needed to disentangle the in-plane and out-of-plane contributions. For fixed excitation fluence, the peak THz field as a function of the pump polarization exhibits a sinusoidal modulation with a periodicity of approximately $\pi$ (Fig. 3c), reflecting the rank-two tensor nature of the photoresponses, which are second order in electric fields. Detailed analysis is included in Supplementary Figs. 5–10. Finally, the TES signal shows a linear dependence on the excitation fluence (Fig. 3d) at low fluences and continues to increase at higher fluences exceeding 60 μJ/cm², albeit with a smaller slope. Such phenomena are likely due to the combined effects of photocurrent saturation due to carrier generation[32] and nonlinearly increasing photocurrents (detailed discussion in Supplementary Note 5).

Figure 3e shows the SHG measurements on the four kinds of CVD-grown samples excited with 800-nm pulses (details in Supplementary Fig. 3). The SHG of several different flakes from each sample was measured to estimate the average intensity and flake-to-flake deviation. 2H MoS₂, 1 $T'$ MoSSe, and 2H MoSSe show high SHG efficiency,

and no SHG signal is detected in 1 $T'$ MoS₂. In 1 $T'$ MoSSe and 2H MoSSe, SHG is further enhanced by a factor of 4 and 3 compared to monolayer 2H MoS₂ respectively, for which high SHG efficiency has been extensively reported[21,34–36]. This highlights the importance of augmented inversion symmetry breaking in Janus structures, which improves even-order nonlinearities. The SHG efficiency in Janus-type samples is further amplified in an angle-resolved SHG measurement (Fig. 3f) that is particularly sensitive to out-of-plane dipoles[25]. In this experiment, the incident angle of the 800-nm fundamental beam deviates from the normal incidence so that the tilted incident beam provides a vertical electric field and interacts with the out-of-plane dipoles effectively. To exclude other geometric factors, an $s$-polarized SHG $I_s$ induced by an in-plane dipole with the same collection efficiency is measured and used to normalize $p$-polarized SHG $I_p$ that contains out-of-plane dipole contribution at non-normal incidence. For 1 $T'$ MoSSe and 2H MoSSe, $I_p/I_s$ symmetrically increases as a function of the incident angle, while 2H MoS₂ shows much smaller angle-dependent changes. This confirms the presence of out-of-plane dipoles in Janus-type samples.

## Theoretical origin of giant terahertz-frequency nonlinearity

The experimental results above indicate that the optical nonlinearity of 1 $T'$ MoSSe can be orders-of-magnitude (e.g., >50 times higher for 18th order HHG; >20 times higher for TES) stronger than those of 2H MoSSe. To understand this effect, we examine the microscopic mechanism underlying the strong THz-frequency nonlinear responses in 1 $T'$ MoSSe. The band structures of 1 $T'$ MoSSe is shown in Fig. 4a. The band inversion of 1 $T'$ MoSSe happens around the $\Gamma$-point. Due to spin-orbit interaction, there is a band reopening at the $\pm\Lambda$-points (marked in Fig. 4a). When the Fermi level is inside the bandgap, the interband transition dipole (Berry connection) $r_{mn}(k) \equiv \langle mk|r|nk \rangle$ plays an essential role in optical processes[37], because it determines the strength of the dipole interaction between electrons and the electric fields. Here

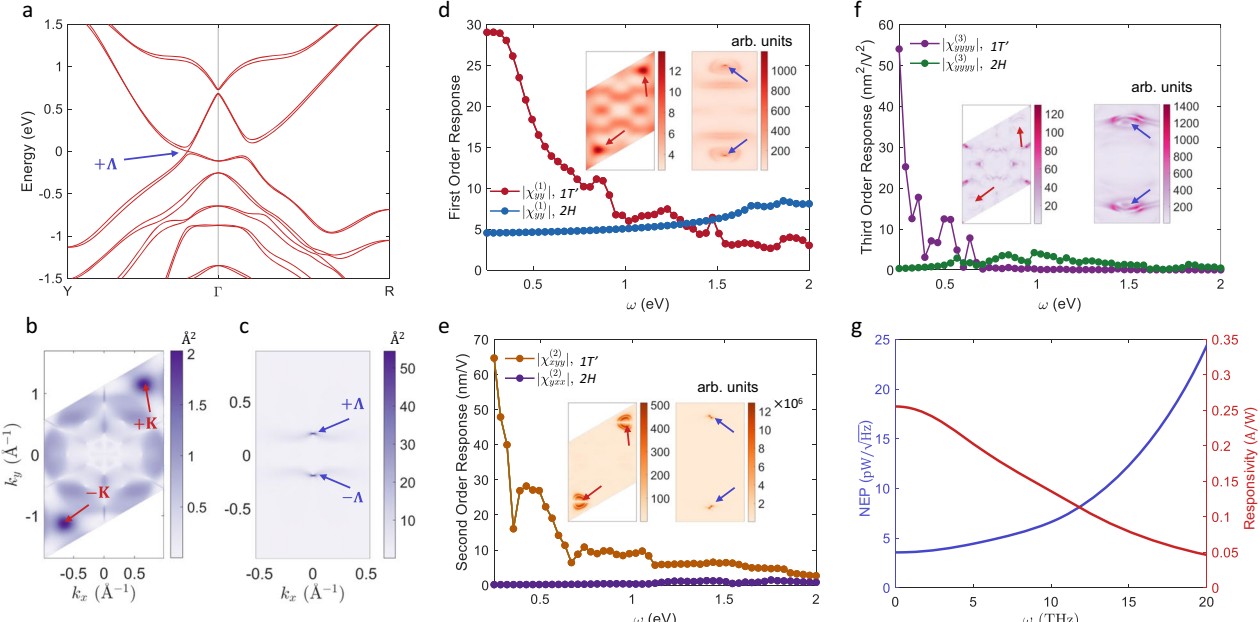

**Fig. 4 | Theoretical calculations of the nonlinear optical response of 2H and 1T′ MoSSe. a** Band structure of 1T′ MoSSe. The energy is offset to the valence band maximum. The band edge of 1T′ MoSSe is located at the Λ point, which is marked in (**a**). **b, c** The magnitude of the interband transition dipole $|r_{vc}(k)|$ for (**b**) 2H and (**c**) 1T′ MoSSe in the first Brillouin zone. The colormap is in the unit of A. **d–f** First, second, and third-order optical responses of 1T′ and 2H MoSSe as a function of incident light frequency ω. Insets of (**d–f**) show the k-resolved contributions (with

arbitrary units) to the total response function at ω = 1 eV for (left) 2H and (right) 1T′ MoSSe. Red (blue) arrows in (**b, c**) and insets of (**d–f**) denote the ±K (±Λ) points, which are the bandedge of 2H (1T′) MoSSe. In insets of (**d–f**), the Brillouin zone is zoomed in around the ±Λ points for 1T′ MoSSe to give better visibility. **g** The NEP (left y axis) and photo-responsivity (right y axis) of 1T′ JTMD THz detector as a function of the light frequency ω.

$r$ is the position operator, while $|mk\rangle$ is the electron wavefunction at band $m$ and wavevector $k$. In Fig. 4b, c, we plot $|r_{vc}(k)|$ of 2H and 1T′ MoSSe, where $v$ ($c$) denotes the highest valence (lowest conduction) band. For 2H MoSSe, the maximum value of $|r_{vc}(k)|$ is around ~2 A near the band-edge (±K points), while for 1T′ MoSSe, $|r_{vc}(k)|$ can reach ~50 A near the band-edge (±Λ points). Consequently, electrons in 1T′ MoSSe would have stronger dipole interaction and hence faster interband transitions under light illumination. This is attributed to the topological enhancement, that is, band inversions in topological 1T′ MoSSe lead to wavefunction hybridization and hence larger wave-function overlap between valence and conduction bands near the band edge, which accelerates the interband transitions[3,38,39]. The calculated first, second, and third-order nonlinear susceptibilities of 2H and 1T′ MoSSe are shown in Fig. 4d–f. For ω ≲ 0.5 eV, the responses of 1T′ MoSSe are significantly stronger than those of 2H MoSSe. For ω ≳ 1 eV, the responses of 1T′ and 2H MoSSe are relatively close, consistent with experimental HHG, TES, and SHG measurements at different wavelengths. In the insets of Fig. 4d–f, we plot the k-resolved contributions $I^{(i)}(k)$ to the optical susceptibility at ω = 1 eV (see "Methods" for details). Notably, the maximum value of $I^{(i)}(k)$ of 1T′ MoSSe, located around the ±Λ points, is larger by orders-of-magnitude than that of 2H MoSSe. This indicates that k-points near the ±Λ points, which are influenced by topological enhancement, make major contributions to the total susceptibility even at ω = 1 eV. Note that the bandgap of 1T′ MoSSe is on the order of 10 meV, and thus interband transitions of electrons near the ±Λ points are far off-resonance with ω = 1 eV photons. However, the contributions near the ±Λ points still dominate those at other k-points where resonant interband transitions could happen. This again suggests the impor-tance of the topological enhancement and the large interband transi-tion dipoles near the ±Λ points. The topological enhancement gradually decays at large ω. Consequently, the optical responses could be stronger in 2H MoSSe with ω ≳ 1 eV. Other in-plane elements of the SHG tensor are shown in Supplementary Fig. 14. We also note that the

theoretical calculations here should only be considered as qualitative estimations, and several issues can lead to inaccuracies. For example, the density functional theory calculations suffer from intrinsic errors regarding some electronic properties, including bandgaps. Some many-body interactions are also ignored in the calculations here. Besides, the theoretical calculations deal with ideal materials, which should be distinguished from the real samples used in experimental that are influenced by doping levels, etc. Future theoretical and experimental developments could yield more accurate information for quantitative theory-experiment comparison.

## Discussion

The giant nonlinearities of 1T′ JTMD, corroborated by both experi-mental and theoretical results above, support the giant THz frequency photocurrent responses of 1T′ JTMDs predicted by theory[3] and pre-lude that 1T′ JTMD could serve as efficient dark-current-free THz detectors via the nonlinear bulk photovoltaic effect[10]. Our calculations indicate that the intrinsic photo-responsivity and noise equivalent power of the 1T′ JTMD THz detector can outperform many current room-temperature THz sensors based on Schottky diodes or silicon field-effect transistors[40,41], albeit lower than the best pyroelectric detectors and bolometers[40] (Fig. 4g, see also Supplementary Note 1 and Supplementary Fig. 11). We foresee stacking multiple monolayer 1T′ JTMDs and using field-enhancement structures[42] can further enhance the responsivity[43] and enable a facile usage of this detector for THz sensing purposes.

In conclusion, we demonstrate giant nonlinear responses in monolayer 1T′ MoSSe, a prototype Janus topological semiconductor. Comparative experiments with different crystal phases (2H vs. 1T′) and symmetry types (Janus vs. non-Janus) indicate that 1T′ MoSSe pos-sesses orders-of-magnitude enhancement in HHG and second-order THz emission efficiency, and a few times enhancement in infrared SHG. Supported by theoretical calculations, our results elucidate that the remarkable enhancements originate from augmented structural

asymmetry in Janus-type structures and topological band-mixing in 1 $T'$ phases. The boosted HHG efficiency and the high fabrication versatility[27] of 1 $T'$ JTMDs prelude a plethora of applications in light-wave electronics[44,45] in the monolayer limit. Meanwhile, the giant THz-frequency nonlinearities observed in this work could enable THz detection[46,47] with a large photo-responsivity at sub-A/W level and noise equivalent power down to the pW/$\sqrt{\text{Hz}}$ level.

## Methods

### Growth of 1 T′ MoS₂ monolayer flakes

The precursor $K_2MoS_4$ was prepared according to the previously reported synthesis procedures[48]. The growth of 1 $T'$ $MoS_2$ monolayer flakes was carried out in a standard CVD furnace with a 1-inch quartz tube under atmospheric pressure. A fresh-cleaved fluorophlogopite mica substrate with $K_2MoS_4$ powders were placed in the center of the furnace. After the system was purged with Ar for 10 min, the furnace was heated up to 750 °C in 40 min with 100 sccm Ar. When the temperature of the furnace reached 750 °C, 10 sccm $H_2$ was introduced and the flow rate of Ar was decreased to 90 sccm. After 5 min, the mica substrate was rapidly pulled out of the furnace heating zone. After cooling down to room temperature, the 1 $T'$ $MoS_2$ monolayer flakes were obtained.

### Synthesis of 1 T′ MoSSe monolayer flakes

The synthesis of monolayer 1 $T'$ MoSSe is realized by a room-temperature atomic layer substitution method[27] from 1 $T'$ $MoS_2$[49]. A remote commercial inductively coupled plasma (ICP) system was used to substitute the top-layer sulfur atoms of monolayer 1 $T'$ $MoS_2$ with selenium. The potassium-assisted CVD-grown monolayer 1 $T'$ $MoS_2$ was placed in the middle of a quartz tube. The plasma coil placed at the upstream of CVD furnace. The distance between the sample and the plasma coil is around 10 cm, with the selenium powder placed on the other side. At the beginning of the process, the whole system was pumped down to a low mTorr to remove air in the chamber. Then, hydrogen was introduced into the system with 10 sccm and the plasma generator was ignited for 20 min. The hydrogen atoms assist the removal of the sulfur atoms on the top layer of $MoS_2$, at the same time, the vaporized selenium filled in the vacancy of the sulfur atoms, resulting in the asymmetric Janus topological structure of MoSSe. The whole process was performed at room temperature. After the reaction, Ar gas was purged into the system with 100 sccm to remove the residual reaction gas, and the pressure was recovered to atmospheric.

### HHG, THz emission, and SHG setups

For HHG, the fundamental laser beam has a wavelength of 5.0 μm with a repetition rate of 1 kHz and a pulse duration of ~70 fs. It is generated by the difference frequency of the signal and idler beam from an optical parametric amplifier pumped with a Ti: Sapphire chirped-pulse amplifier (6 mJ, 1 kHz). The fundamental MIR beam with a 10 μJ pulse energy was focused on the sample with a ZnSe lens with a focal length of 15 cm. The measurements are performed in a transmission geometry at normal incidence. Generated HHG signals transmitted through the sample are collected by a CaF₂ lens and directed and dispersed in a spectrometer (Princeton Instruments HRS-300) and detected by a charge-coupled device (CCD) camera (Princeton Instruments PIXIS 400B). We note that the distorted shapes of high-order harmonics are due to chromatic aberration when focusing and imaging the HHG from the entrance slit of the spectrometer to the CCD camera.

For THz emission measurement, the ultrafast laser excitation was provided by a mode-locked Ti:Sapphire laser with pulses of 40-fs (FWHM) duration and 5.12-MHz repetition rate. After focusing the laser beam on the sample, the refocused THz radiation from the sample was detected using the electrooptic (EO) effect in a non-centrosymmetric crystal (1-mm-thick ZnTe or 258-μm-thick GaP). The induced

birefringence in the EO crystal was recorded at different delay times by a laser probe pulse passing through a polarizing beamsplitter (Wollaston prism) and impinging on a balanced photodetector. The power imbalance was fed into a lock-in amplifier synchronized with modulation of the excitation beam at 320 kHz by an acousto-optic modulator. By scanning the time delay between the excitation and probe pulses, the temporal profile of the transient THz electric field could be mapped. A pair of wire-grid polarizers were used to determine the THz emission polarization.

For SHG measurement, the fundamental pulses are provided by a mode-locked Ti:Sapphire oscillator at 800-nm wavelength, 5.12-MHz repetition rate, and 40-fs pulse duration. They are focused on the sample with a ×20 objective, and the generated SHG light from a single flake is filtered and detected by a photomultiplier tube. For out-of-plane measurement, a collimated $p$-polarized pump beam with a 1 mm spot size is guided to the objective back aperture (D = 7.6 mm). The beam was focused at the sample with a tilted angle and generated an oscillating vertical electrical field to drive the out-of-plane dipole for SHG. The SHG (green) is collected with the same objective and analyzed by a polarizer. The beam position at the objective back aperture can be scanned along the $x$-direction with a motorized stage, which tunes the incident angle accordingly.

### Ab initio calculations

The ab initio density functional theory (DFT)[50,51] calculations are performed using the Vienna ab initio simulation package (VASP)[52,53]. The exchange-correlation interactions are included using generalized gradient approximation (GGA) in the form of Perdew–Burke–Ernzerhof (PBE)[54]. Core and valence electrons are respectively treated by projector augmented wave (PAW) method[55] and plane-wave basis functions. The first Brillouin zone is sampled by a $13 \times 17 \times 1$ and $17 \times 17 \times 1$ $k$-mesh for 1 $T'$ and $2H$ structures, respectively.

### Nonlinear optical susceptibility calculations

After the DFT results are obtained, a tight-binding (TB) Hamiltonian in the Wannier basis is built using the Wannier90 package[56]. The TB Hamiltonian is then used to interpolate the relevant properties on a denser $k$-mesh.

The first-order susceptibility is calculated within the velocity gauge

$$\chi_{ij}^{(1)}(\omega;\omega) = -\frac{e^2}{\varepsilon_0 \omega} \int \frac{d^3\boldsymbol{k}}{(2\pi)^3} \sum_{mn} \frac{f_{mn}}{E_{mn}} \frac{v_{nm}^i v_{mn}^j}{E_{mn} - \hbar\omega} \tag{1}$$

Here $m,n$ label the electron states, while $v_{mn}^i \equiv \langle m|v^i|n\rangle$ is the velocity operator. $E_{mn}$ and $f_{mn}$ are respectively the difference in energy and occupation number between $|m\rangle$ and $|n\rangle$. $\varepsilon_0$ is the vacuum permittivity. The second-order susceptibility is calculated within the length gauge[57]

$$\chi_{ijk}^{(2)}(2\omega;\omega,\omega) = \zeta_{ijk}^{ll}(2\omega;\omega,\omega) + \eta_{ijk}^{ll}(2\omega;\omega,\omega) + \sigma_{ijk}^{ll}(2\omega;\omega,\omega) \tag{2}$$

where

$$\zeta_{ijk}^{ll}(2\omega;\omega,\omega) = \frac{e^3}{\varepsilon_0} \int \frac{d^3\boldsymbol{k}}{(2\pi)^3} \sum_{mnl} \frac{r_{nm}^i r_{ml}^j r_{ln}^k}{E_{ln} - E_{ml}} \left( \frac{f_{ml}}{E_{ml} - \hbar\omega} + \frac{f_{ln}}{E_{ln} - \hbar\omega} + \frac{2f_{nm}}{E_{mn} - \hbar\omega} \right) \tag{3}$$

$$\eta_{ijk}^{ll}(2\omega;\omega,\omega) = \frac{e^3}{\varepsilon_0} \int \frac{d^3\boldsymbol{k}}{(2\pi)^3} \left\{ \sum_{mnl} E_{mn} r_{nm}^i \left\{ r_{ml}^j r_{ln}^k \right\} \left[ \frac{f_{nl}}{E_{nl}^2(E_{ln} - \hbar\omega)} + \frac{f_{lm}}{E_{lm}^2(E_{ml} - \hbar\omega)} \right] \right.$$
$$\left. -8i\hbar \sum_{nm} \frac{f_{nm} r_{nm}^i}{E_{mn}^2(E_{mn} - 2\hbar\omega)} \left\{ \Delta_{mn}^j r_{mn}^k \right\} - 2\sum_{nml} f_{nm} r_{nm}^i \frac{\left\{ r_{ml}^j r_{ln}^k \right\}(\omega_{ln} - \omega_{ml})}{E_{mn}^2(E_{mn} - 2\hbar\omega)} \right\} \tag{4}$$

$$\sigma_{ijk}^{ll}(2\omega;\omega,\omega) = \frac{e^3}{2\varepsilon_0}\int\frac{d^3\boldsymbol{k}}{(2\pi)^3}\left\{\sum_{nml}\frac{f_{nm}}{E_{mn}^2(E_{mn}-\hbar\omega)}\left[E_{nl}r_{lm}^i\left\{r_{mn}^j r_{nl}^k\right\}\right.\right.$$
$$\left.\left.- E_{lm}r_{nl}^i\left\{r_{lm}^j r_{mn}^k\right\}\right] + i\hbar\sum_{nm}\frac{f_{nm}r_{nm}^i}{E_{mn}^2(E_{mn}-2\hbar\omega)}\left\{\Delta_{mn}^j r_{mn}^k\right\}\right\} \tag{5}$$

Here the inter-band position matrix is $r_{mn}^i = \frac{\hbar v_{mn}^i}{iE_{mn}}$ for $m\neq n$ and $r_{mn}^i = 0$ for $m = n$. $\Delta_{mn}^i \equiv v_{mm}^i - v_{nn}^i$ is the difference in band velocities. Meanwhile, for two numbers $A$ and $B$ one has $\{AB\} \equiv \frac{1}{2}(AB+BA)$.

The ab initio theory for calculating the third-order susceptibility is not well-developed. Here we use the velocity gauge formula[58]

$$\chi_{ijkl}^{(3)}(3\omega;\omega,\omega,\omega) = \frac{ie^4}{4\varepsilon_0\omega^3}\int\frac{d^3\boldsymbol{k}}{(2\pi)^3}\sum_{nmpq}\frac{v_{nm}^i}{E_{mn}-3\hbar\omega}$$
$$\left[\frac{v_{mp}^j\left(g_{qn}^l v_{pq}^k - g_{pq}^l v_{qn}^k\right)}{E_{pn}-2\hbar\omega} + \frac{v_{qn}^j\left(g_{mp}^l v_{pq}^k - g_{pq}^l v_{mp}^k\right)}{E_{mq}-2\hbar\omega}\right] \tag{6}$$

Here $g_{mn}^i \equiv \frac{f_{nm}v_{mn}^i}{E_{nm}-\hbar\omega}$. Equation (6) experiences a spurious divergence in the real part of $\chi_{ijkl}^{(3)}$. Therefore, we first calculate the imaginary part of $\chi_{ijkl}^{(3)}$, and then obtain the real part from the Kramers–Kronig relations[59].

The $\boldsymbol{k}$-resolved contributions to the total susceptibility $I^{(i)}(\boldsymbol{k})$, which are shown in the inset of Fig. 4b–d, are defined as the integrand of the Brillouin zone integration in Eqs. (1, 2, 6). The Brillouin zone integrations is performed by a $\boldsymbol{k}$-mesh sampling, $\int\frac{d^3\boldsymbol{k}}{(2\pi)^3} = \frac{1}{V}\sum_{\boldsymbol{k}}w_{\boldsymbol{k}}$, where $V$ is the volume of the unit cell and $w_{\boldsymbol{k}}$ is the weight factor. Since 2D materials do not have well-defined volume, we use $V = Sl_{\text{eff}}$, where $S$ is the area of the 2D unit cell, while $l_{\text{eff}}$ is taken as 6 Å for all materials. The convergence in the $\boldsymbol{k}$-mesh is tested.

## Data availability
Relevant data supporting the key findings of this study are available within the article and the Supplementary Information file. All raw data generated during the current study are available from the corresponding authors upon request.

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

## Acknowledgements

This work was primarily funded through the Department of Energy, Office of Basic Energy Sciences, Division of Materials Sciences and Engineering, under contract DE-AC02-76SF00515. The high harmonic generation experiments were supported by the US Department of Energy, Office of Science, Basic Energy Sciences, Chemical Sciences, Geosciences, and Biosciences Division through the AMOS program. Y.G. acknowledges the financial support from Zhejiang University. H.X. and J.L. were supported by an Office of Naval Research MURI through grant #N00014-17-1-2661. E.S. and H.S. acknowledge the financial support from Research Center for Industries of the Future at Westlake University, National Natural Science Foundation of China (grant no. 52272164). J.K. and T.Z. acknowledge the financial support from US Department of Energy (DOE), Office of Science, Basic Energy Sciences under Award DE-SC0020042.

## Author contributions

J.S., H.X., Y.G. and A.L. designed the study; Y.G. performed the Janus material synthesis and Raman characterization; H.X. and J.L. performed the theoretical analyses and ab initio calculations.; C.H. and J.S. performed HHG measurements under the supervision of S.G. and A.L.; C.X. and J.S. performed TES measurements under the supervision of A.L.; C.H.F. synthesized typical transition metal dichalcogenides under the supervision of L.J.; J.S., F.Q. and L.Y. performed SHG measurements under the supervision of A.L. and T.H.; A.J. synthesized wafer-scale 2H $MoS_2$ under the supervision of F.L.; H.S., T.Z., E.S. and J.K. participated in data analysis; J.S., H.X. and Y.G. wrote the manuscript; All authors read and revised the manuscript.

## Competing interests

The authors declare no competing interests.
