## [Peer Review File · Nature Communications]

Giant room-temperature nonlinearities from a monolayer Janus topological semiconductorREVIEWER COMMENTS

Reviewer #1 (Remarks to the Author):

Shi et al. study the nonlinear response of MoSSe and MoS₂ in different phases, topological and trivial. They use three different methods, high-order harmonic generation (HHG), THz spectroscopy, and second-harmonic generation. The two latter were performed using 800-nm incident light, the HHG with 5 microns. The experimental results show huge nonlinearities in a wide range of emission frequencies for the topological monolayer 1T' MoSSe. Theoretical analysis shows that asymmetries and topological band inversion help to enhance the dipole transition matrix elements. The work is very impressive, especially on the experimental side, and I recommend publication in Nature Communications. I have only a few points the authors should address prior to publication:

1. From the band structure in Fig. 4a, I infer that the photon energy corresponding to 5 microns is larger than the inverted band gap. Is this so? That would be quite unusual HHG. The authors might clarify how their incident photon energy and that of the emitted HHG photons compare to the band gap.
2. Also concerning HHG: Can the authors say how far the electrons are accelerated away from the Lambda point?
3. The authors probably mean "so far" in line 57.
4. Is the topological non-triviality necessary to achieve such large nonlinearities? Asymmetry and large transition matrix elements due to hybridisation might occur in topologically trivial materials as well. In one of the subsection headings the author claim that topological bands "boost" the HHG efficiency. Is that generally true or is that special to 1T' MoSSe?
5. If the laser spot size is larger than the flakes, do topologically protected edge currents play a role? That would be exciting.
6. Are the same samples used for the three different kinds of experiments?
7. Please add the band gaps in Table 1.
8. In the Conclusions, the authors, like others, advertise "on-chip attosecond photonics" and "light-wave electronics". I wonder how attoseconds come into play at such long wavelengths. Concerning "light-wave electronics", miniaturisation is a problem because of the long wavelengths and large spot sizes. Or do the authors have some other ideas, e.g., employing near fields?

Reviewer #2 (Remarks to the Author):

Lindenberg and co-workers have investigated nonlinear optical responses from two-dimensional topological semiconductors. In this manuscript, it has been shown that the 1T' phase of the Janus MoSSe, a modified two-dimensional semiconductor derived from MoS₂, exhibits enhanced nonlinear optical signals, such as high-harmonic generation, terahertz emission, and second harmonic generation. The enhanced nonlinearity at room temperature is attributed to the absence of inversion symmetry and the presence of topological band mixing. The present manuscript, in its current form, is not suited for Nature Communications. Here are my concerns:

1. It has been shown that the two-dimensional monolayer semiconductor exhibits higher efficiency of harmonic generation in comparison to its bulk counterpart, see Ref. [27]: Nature Physics 13, 262 (2017). Then, I fail to understand the motivation to compare and discuss the present results with ZnO on page 5. This is redundant. Authors could remove the such discussion. It is crucial to show how the nonlinearities from the 1T' phases of MoSSe and MoS₂ scale with electron density (carrier density) and the number of electrons per unit cell.

2. Could the authors discuss the damage thresholds of the Janus MoSSe with its counterparts MoS2 in both the 1T' and 2H phases? This is crucial to establish the relative superiority of Janus MoSSe.
3. Are the results shown in Fig. 2 obtained after normalizing the first harmonic in 1T' MoSSe and 2H MoS2? I think the authors could make a fair comparison in both cases after normalizing the intensity (yield) of the first harmonic. Also, why the higher-order harmonics are chirped (distorted) at higher energy in Fig. 2(a), please discuss.
4. In Fig. S4, I can only see some of the flakes have sparse spacing, as written in lines 112-113. Please discuss it.
5. Discussion related to Fig. 3(c): How the crystal symmetry can be accessed from the polarization?
6. I don't see how the THz emission gets saturated at higher fluence in Fig. 3(d). I could see the signal monotonically increasing with the fluence and not saturated. This requires a justification.
7. A theoretical discussion is made to support the findings of giant THz nonlinearity in a hand-waving way based on considerable significant value interband dipole amplitude. I didn't understand why the authors provided a theoretical discussion of the THz nonlinearity and not the HHG shown in Fig. 2. I am not sure how the topological nature of Janus MoSSe enhances the nonlinear signal of HHG. I strongly suggest numerical simulations to support their findings. Authors could use tight-binding-based simulations to have a better understanding.
8. About Table 1: comparing present results with gas and liquid is irrelevant. I suggest making a comparison with solid samples is sufficient
9. About references: The authors have mostly referenced their own work, and some meaningful references are missing. For better readability, I suggest providing a few missing references, such as how the present work could be related to clean energy harvesting in line 78 or related to building blocks for scalable attosecond sources in the abstract. Moreover, light-driven nonlinearities have been used to realize valleytronics in graphene [Optica 8, 422-427 (2021)], the relation of the anomalous HHG to anomalous nonlinear Hall effect [Phys. Rev. B 105, 155140 (2022)]. Thus, it is crucial to refer to these works.

After incorporating suggested changes and improving the discussion, the present manuscript could be recommended for publication in Nature Communications.

Reviewer #3 (Remarks to the Author):

The manuscript by J. Shi and H. Xu et al. reports strongly enhanced nonlinear optical responses in monolayer Janus-type topological semiconductors. The study combines a systematic comparison of four different transition metal dichalcogenides by three state of the art nonlinear spectroscopy techniques: THz emission spectroscopy (TES) and second harmonic generation (SHG) probe second order nonlinear effects in opposite frequency regimes, whereas high harmonics generation (HHG) is employed to investigate higher order nonlinearities in the mid-infrared (MIR). Based on the MoS2 family, the four investigated monolayers of 2D van der Waals materials are distinct by their presence or lack of topological band mixing (1T' vs. 2H structures) and degree of inversion symmetry breaking (top layer S -> Se substitution for Janus-type compounds). The Janus 2D topological insulator (TI) 1T' MoSSe combines topological band properties with strong inversion symmetry breaking and therefore exhibits extremely enhanced TES and HHG, whereas its visible SHG is on the same order of magnitude as its topologically trivial 2H sibling. By additional variation of several experimental parameters and based on theory modelling, the authors thus find a drastic enhancement of the nonlinear susceptibilities for decreasing photon energies from 1 eV down to the THz frequency range; likely connected to the interband transition close to the new λ

high symmetry points.

The research topic of monolayer 2D materials with THz and MIR optical nonlinearities is timely and highly relevant to contemporary material science and potential future application for detector or emitter devices. The manuscript is well written and concise. The vast amount of experimental and simulation data is presented in a clear and comprehensible way. Sample preparation and the variety of state of the art methods are clearly not trivial and conducted in a reliable way. The main conclusions of the paper concerning a greatly enhanced nonlinear optic responses in the Janus-type topological compound 1T' MoSSe is plausible. Nevertheless, the manuscript could benefit from a clearer discussion on the influences of the four materials' distinct properties, e.g. by separating the role of the topological band texture and degree of inversion symmetry breaking. Moreover, the different techniques and the respective order of nonlinearities could be better explained in terms of their frequency realms: e.g. the $\chi^{(2)}$ of TES connecting NIR/VIS and THz wavelengths versus SHG depending on NIR/VIS and close-to-UV wavelengths. How do these frequency combinations relate to the bandgaps of the investigated materials?

Overall, I believe that the work is suitable for a publication as an article in Nature Communications if the following minor remarks are addressed:

A) Concerning the HHG experiments in Fig. 2: Why is the even order HHG stronger in 2H than in 1T' MoS₂, but opposite for Janus-type MoSSe? What do the asterisks on the HHG spectra indicate?

B) Concerning the THz emission: The authors conclude primarily in-plane contributions from the THz emission polarization analysis in Fig. 3b. Is this conclusion justified when the x-polarization with in- and out-of plane contribution dominates the y-polarized component by about a factor of 3? Moreover, was the THz peak field as a function of the pump polarization (Fig. 3c) measured for a specific emitted THz polarization (e.g. only for the E_x or E_y projections)? Or is it really the peak field independent of its polarization?

C) Comparing TES and SHG both relying on $\chi^{(2)}$ but with different frequency relations: Why is there significant SHG for 2H MoS₂ and MoSSe, but vanishing optical rectification leading to basically zero THz emission? A discussion on these subtle differences would be surely helpful.

D) According to Fig. 3f, the incident angle dependence seems to be dominated by the Janus structure with a minor influence of the topological properties. Is there a simple reason for that?

E) According to the calculated second order nonlinear susceptibility of 2H MoSSe in Fig. 4e, the SHG should be nearly vanishing, which is in stark contrast to the experimental observation in Fig. 3e. Where does this contradiction come from? Is it related to different tensor elements not shown in Fig. 4e? Why did the authors choose to calculate this specific tensor elements shown in Fig. 4?

F) The detector discussion in the end of the manuscript seems to come out of nowhere. In the main text, this should be better motivated in relation to the experimental results. Furthermore, the NEP abbreviation for noise-equivalent power is not defined in the main manuscript. For the broad audience of the targeted journal, the meaning of Fig. 4g should be also better explained.

Reply to Reviewers

We thank all reviewers for their detailed comments which have improved the paper. In the following we respond to all comments and detail the changes made to the manuscript, with our responses shown in red.

Reviewer #1

Shi et al. study the nonlinear response of MoSSe and MoS₂ in different phases, topological and trivial. They use three different methods, high-order harmonic generation (HHG), THz spectroscopy, and second-harmonic generation. The two latter were performed using 800-nm incident light, the HHG with 5 microns. The experimental results show huge nonlinearities in a wide range of emission frequencies for the topological monolayer 1T' MoSSe. Theoretical analysis shows that asymmetries and topological band inversion help to enhance the dipole transition matrix elements. The work is very impressive, especially on the experimental side, and I recommend publication in Nature Communications.

We thank the Referee for their careful reading of our manuscript and the encouraging comments. We have carefully revised our manuscript according to the critical comments below.

I have only a few points the authors should address prior to publication:

1. From the band structure in Fig. 4a, I infer that the photon energy corresponding to 5 microns is larger than the inverted band gap. Is this so? That would be quite unusual HHG. The authors might clarify how their incident photon energy and that of the emitted HHG photons compare to the band gap.

That Referee is right. 5 microns corresponds to 0.25 eV and is above the bandgap of 1T' MoSSe and MoS₂, and is below the bandgap of 2H structures. We note that efficient HHG has been observed in various classes of materials excited by light above the bulk (or surface) bandgap. For example, ZnO [Ref 26: *Nat. Phys.* **7**, 138-141 (2011)], Bi₂Se₃ [Ref. 14: *Nature* **593**, 385–390 (2021); Ref 15: *Nat. Photon.* **16**, 620-624 (2022); Ref 16: *Nat. Phys.* **17**, 311-315 (2021)], graphene [Ref 18: *Nature* **561**, 507-511 (2018); *Science* **356**, 736-738 (2017)], etc. We have added a few lines in the main text and a column of bandgap values in Table 1 to clarify this aspect.

2. Also concerning HHG: Can the authors say how far the electrons are accelerated away from the Lambda point?

The Referee raised an important question. We can possibly estimate the distance using the semi-classical equation of motion of the electrons: $\frac{\partial k}{\partial t} = \frac{eE}{\hbar}$, where E is the electric field [*Nature* **550**, 224–228 (2017)]. This equation yields $\Delta k \approx \frac{\tau eE}{\hbar}$, where τ is the decoherence time of the electrons. Taking $\tau = 1$ fs and $E = 1$ V/nm, one has $\Delta k \approx 0.15 \text{ \AA}^{-1}$, which is 1 ~ 10 % of the size of the Brillouin zone. This is at similar order of magnitude compared with literature [*Nature* **616**, 696–701 (2023)]. We have added this estimation to the Supplementary Note 3.

3. The authors probably mean “so far” in line 57.

The Referee is correct. This is now revised.

4. Is the topological non-triviality necessary to achieve such large nonlinearities? Asymmetry and large transition matrix elements due to hybridization might occur in topologically trivial materials as well. In one of the subsection headings the authors claim that topological bands “boost” the HHG efficiency. Is that generally true or is that special to 1T’ MoSSe?

We thank the reviewer for these insightful comments. First, we agree with the reviewer that large asymmetry can lead to enhanced nonlinear optical responses, especially for even-order responses. This is generally true and holds in topologically trivial materials. On the other hand, nontrivial topology can contribute to an additional enhancement in the nonlinear optical responses, as the band inversion leads to stronger wavefunction hybridization and larger transition matrix elements. Detailed discussions on the topological enhancement in first- and second-order responses can be found in [*J. Phys. Chem. Lett.* **11**, 6119 (2020)] and [*npj Comput. Mater.* **7**, 31 (2021)], respectively.

Here, we further demonstrate the topological enhancement in high harmonic generation (HHG). We adopt the Su–Schrieffer–Heeger (SSH) model, which is a minimal and prototypical model that exhibits a topological phase transition. The SSH model is a one-dimensional lattice model, and each unit cell contains two atoms labelled with A and B , respectively. The Hamiltonian of the SSH can be expressed as

$$H = v \sum_n |n, B\rangle\langle n, A| + w \sum_n |n + 1, A\rangle\langle n, B| + h. c. \quad (s)$$

where $h. c.$ stands for Hermitian conjugation. $|n, A(B)\rangle$ is the orbital on the A (B) atom in the n -th unit cell. Meanwhile, v and w are two different hopping amplitudes. Notably, the SSH model undergoes a topological phase transition at $v = w$, and is a topological (normal) insulator when $v < w$ ($v > w$).

We can compare the HHG responses of the SSH model in topological and normal phases. Specifically, we use two sets of parameters ($v = 0.8$ eV, $w = 1$ eV) and ($w = 1$ eV, $v = 0.8$ eV), whereby the SSH model is a topological insulator (TI) and normal insulator (NI), respectively. The band structures in these two phases are identical (Figure R1a). However, the transition dipole is larger in the TI phase due to the band inversion (Figure R1b), which leads to stronger HHG shown in Figure R1c and R1d. One can see that the HHG is enhanced by around $3 \sim 4$ times in the TI phase, even though the band structure of the TI phase is identical to that in the NI phase.

The results above based on generic tight-binding model suggest that the topological enhancement in nonlinear optical responses should be generally true, and is not specific to 1T’ MoSSe. We have added this new calculation to the main text. Details on the calculation of the HHG in the SSH model has been added to the Supplementary Note 4.

Figure R1. HHG of the SSH model. (a) Band structure of the SSH model with different sets of parameters, which corresponds to a TI and a NI, respectively. (b) Interband transition dipole in the SSH model in TI and NI phases. (c) Induced electric field under a femto-second pumping laser. (d) HHG signal of the SSH model in TI and NI phases. Parameters of the SSH model and the pumping lasers are described in detail in the Supplementary Information.

5. If the laser spot size is larger than the flakes, do topologically protected edge currents play a role? That would be exciting.

We thank the reviewer for this insightful comment. Yes, topologically protected edge currents can potentially play a role. Electrons do not suffer from back-scatterings on the edge, which improves the electron coherence and hence enhance the nonlinear optical responses. Actually, similar effects on the surface state of a three-dimensional topological insulators Bi_2Se_3 have been studied [*Nat. Photon.* **16**, 620 (2022)]. We agree it would be exciting if the topologically protected edge state enhances HHG. Although there is discussion and evidence for such an effect [*Phys. Rev. Lett.* **120**, 177401 (2018); *Phys. Rev. B* **106**, 054303 (2022)], future work is needed to prove its existence in our case.

6. Are the same samples used for the three different kinds of experiments?

We thank the reviewer for mentioning this. We have measured four different types of samples at four different phases (1T' MoS_2 , 2H MoS_2 , 1T' MoSSe , 2H MoSSe). It is not the same flake, but we randomly chose different flakes for comparison. We also showed the statistics in Fig. 3h. That means the enhancement is not specific to a single flake.

7. Please add the band gaps in Table 1.

We have added the bandgaps in Table 1.

8. In the Conclusions, the authors, like others, advertise “on-chip attosecond photonics” and “light-wave electronics”. I wonder how attoseconds come into play at such long wavelengths. Concerning “light-wave electronics”, miniaturization is a problem because of the long wavelengths and large spot sizes. Or do the authors have some other ideas, e.g., employing near-fields?

We appreciate the Referee for these further thoughts. The highest harmonic we can generate is currently limited by detection optics and it is conceivable that we can generate even higher harmonics (i.e., shorter wavelength, larger bandwidth) to potentially enable attosecond applications. But we agree with the Referee that there is a lot of additional work needed to realize attosecond applications, for example the miniaturization problem. And the Referee is right that this problem can be possibly tackled with metamaterial, waveguide or near-field local enhancement operating at mid-infrared regime [*Adv. Mater.* **24**, OP98-OP120 (2012); *Nanophotonics* **7**, 393–420 (2018)]. Since it is not directly relevant to the main message in the current work, we have deleted the discussions on the implication in “on-chip attosecond photonics”.

Reviewer #2

Lindenberg and co-workers have investigated nonlinear optical responses from two-dimensional topological semiconductors. In this manuscript, it has been shown that the 1T' phase of the Janus MoSSe, a modified two-dimensional semiconductor derived from MoS₂, exhibits enhanced nonlinear optical signals, such as high-harmonic generation, terahertz emission, and second harmonic generation. The enhanced nonlinearity at room temperature is attributed to the absence of inversion symmetry and the presence of topological band mixing. The present manuscript, in its current form, is not suited for Nature Communications.

We thank the Referee for the careful reading and for affirming the significance of our work. Below we address the Reviewer's questions and highlight the changes we made to our manuscript to account for their comments. We believe our manuscript has been significantly improved and should be suitable for Nature Communications.

Here are my concerns:

1. It has been shown that the two-dimensional monolayer semiconductor exhibits higher efficiency of harmonic generation in comparison to its bulk counterpart, see Ref. [27]: *Nature Physics* **13**, 262 (2017). Then, I fail to understand the motivation to compare and discuss the present results with ZnO on page 5. This is redundant. Authors could remove such discussion. It is crucial to show the nonlinearities from the 1T' phases of MoSSe and MoS₂ scale with electron density (carrier density) and the number of electrons per unit cell.

We thank the Referee for pointing this out. We agree with the Referee that Ref [*Nat. Phys.* **13**, 262 (2017)] has shown that HHG in monolayer TMD is more efficient than in bulk TMD, but it did not compare with HHG from other bulk materials (e.g., the prototypical material ZnO) and is complicated by excitonic contributions and quantum confinement effects. Thus there is still value in comparing with ZnO, albeit not directly related to the main conclusion. We have moved the discussion on this to Supplementary Note 2.

We appreciate the Referee for mentioning the role of electron density. Nonlinear optical responses, including HHG, have a complicated relationship with carrier density. When the Fermi level resides in the bandgap (no free carriers), then there is only the interband contribution to the responses, namely, the electrons must undergo interband transitions to contribute to the optical responses. On the other hand, if the Fermi level resides in the valence or conduction band (net hole or electron

carriers), then there would be additional intraband processes that contribute to the optical responses. The relationship between NLO responses and the Fermi level has been studied in many previous works, such as Fig. 6a in [*npj Comput. Mater.* **7**, 31 (2021)], Fig. 3 in [*Phys. Rev. B* **97**, 241118(R) (2018)], etc. Generally, the relationship is non-trivial and system-dependent.

2. Could the authors discuss the damage thresholds of the Janus MoSSe with its counterparts MoS₂ in both the 1T' and 2H phases? This is crucial to establish the relative superiority of Janus MoSSe.

We thank the Referee for this important question. From the energetic perspective, the 2H phases should be more stable than the 1T' phases, as reported in Fig. S7 in the Supplementary of Ref [*Science*, **346**, 1344 (2014)]. The 1T' phase is a metastable state, and its energy per unit cell is around 0.5 eV higher than that of the 2H phase. From the practical perspective, the Janus structure is less stable than the non-Janus structure thermodynamically. For its stability under MIR excitation, we did not observe any evident sample damage in either 1T' or 2H phase MoSSe under the highest possible MIR fluence available in our lab. Along with the observation of dramatically enhanced nonlinear responses, they show the superiority of Janus MoSSe. To further clarify, we added a few sentences in Supplementary Note 5 to discuss the sample damage threshold estimation.

3. Are the results shown in Fig. 2 obtained after normalizing the first harmonic in 1T' MoSSe and 2H MoS₂? I think the authors could make a fair comparison in both cases after normalizing the intensity (yield) of the first harmonic. Also, why the higher-order harmonics are chirped (distorted) at higher energy in Fig. 2(a), please discuss.

We thank the Referee for this comment. As for the normalization, however, we do not fully understand the question as we are unsure what the Referee refers to by “first harmonics”. We believe the appropriate procedure would be measuring absolute HHG intensities without normalization at the same pumping condition because the signals at any harmonic orders contain material-specific information under illumination. The comparison should be made by normalizing the intensity of the pumping field, as we performed in the current work by using the same MIR pumping field intensity for all different materials.

The distorted shapes of high-order harmonics are likely due to chromatic aberration instead of chirp when focusing and imaging the HHG from the entrance slit of the spectrometer to the CCD camera. We have added a few sentences in Methods section to clarify this point.

4. In Fig. S4, I can only see some of the flakes have sparse spacing, as written in lines 112-113. Please discuss it.

We thank the Referee for the careful read of the data. We agree that the spacing between the flakes are randomly distributed. Experimentally, we irradiated a single flake in a sparsely populated area and confirmed that we did not experience any interference from neighboring flakes. We achieved this by measuring the polarization dependence and observing the near-perfect cancellation of HHG at specific angles to one of the crystallographic axes. We have added a few lines in the caption of Fig. S4 to further clarify.

5. Discussion related to Fig. 3(c): How the crystal symmetry can be accessed from the polarization?

We thank the Referee for this helpful question. The sinusoidal angle dependence with a period of π shown in Figure 3c is a typical feature of second-order nonlinear optical responses. This is because the total response is $R \sim E \cdot \chi^{(2)} \cdot E$, where $\chi^{(2)}$ is a rank-2 tensor. If the polarization of E is rotated in the x - y plane with an angle of θ , then one has $R(\theta) = E^2 [\chi_{xx}^{(2)} \cos^2 \theta +$

$\chi_{yy}^{(2)} \sin^2 \theta + 2\chi_{xy}^{(2)} \sin \theta \cos \theta$]. Hence, the sinusoidal angle dependence appears as long as $\chi^{(2)}$ is not proportional to identity tensor. From such angle dependence, one can assess e.g., if there is a mirror symmetry. But usually, one cannot get full information on the crystal symmetry (e.g., space group) from these angle dependences. Similar angle dependence data and microscopic origin above has been reported in Ref. 33 [*Sci. Adv.* **5**, 2 (2019)]. To account for the Referee's comments, we have added a few lines in the Supplementary Note 6.

6. I don't see how the THz emission gets saturated at higher fluence in Fig. 3(d). I could see the signal monotonically increasing with the fluence and not saturated. This requires a justification.

This comment by the Referee made us realize that we may have lacked clarity in describing the features characterizing the fluence dependence of THz emission signal. Indeed, the signal continues to increase at higher fluences, albeit with a smaller gradient. Such phenomena are likely due to two combined effects: (1) Saturation of photocurrents due to carrier generation and build up of internal fields, as reported in Ref 32 [*Sci. Adv.* **5**, 2 (2019)]; (2) Nonlinear photocurrents that leads to increasing THz emission signal as a function of excitation fields. We have revised the main text and added a few lines in the Supplementary Note 6 to clarify.

7. A theoretical discussion is made to support the findings of giant THz nonlinearity in a hand-waving way based on considerable significant value interband dipole amplitude. I didn't understand why the authors provided a theoretical discussion of the THz nonlinearity and not the HHG shown in Fig. 2. I am not sure how the topological nature of Janus MoSSe enhances the nonlinear signal of HHG. I strongly suggest numerical simulations to support their findings. Authors could use tight-binding-based simulations to have a better understanding.

We thank the Referee for this critical comment. The reason we did not show HHG in Figure 4 is that the HHG calculation for 1T' MoSSe takes a formidably high computational time. The simulation of HHG involves solving the Semiconductor Bloch equations (SBE), and the computational cost is proportional to $N_b^2 N_k^d$ [*PRA* **101**, 053411 (2020)], where N_b is the number of electronic bands, d is the dimension of the system, while N_k is the number of k -points along each dimension. To the best of our knowledge, the simulations on HHG in two-dimensional materials is restricted to simple tight-binding models with $N_b \lesssim 5$ [see e.g., *Nature* **523**, 572 (2015), *Phys. Rev. A* **103**, 023101 (2021)]. However, the tight-binding model for 1T' MoSSe has $N_b = 44$, and the computation cost is almost 100 times larger and is unaffordable.

To address this comment, in the revised manuscript we present a two-band ($N_b = 2$) Su-Schrieffer-Heeger model to demonstrate the topological enhancement in HHG, which is shown in Figure R1. The results confirm that the topological enhancement persists in higher order responses. To avoid repetition, we refer the Referee to pages 3 and 4 of this document for a detailed description of the new calculation. And we have added this calculation results and details in the main text and Supplementary Note 4.

8. About Table 1: comparing present results with gas and liquid is irrelevant. I suggest making a comparison with solid samples is sufficient

We thank the Referee for the suggestion and have removed the comparison with gas and liquid.

9. About references: The authors have mostly referenced their own work, and some meaningful references are missing. For better readability, I suggest providing a few missing references, such as how the present work could be related to clean energy harvesting in line 78 or related to building

blocks for scalable attosecond sources in the abstract. Moreover, light-driven nonlinearities have been used to realize valleytronics in graphene [Optica 8, 422-427 (2021)], the relation of the anomalous HHG to anomalous nonlinear Hall effect [Phys. Rev. B 105, 155140 (2022)]. Thus, it is crucial to refer to these works.

We are grateful to the Referee for the suggestions. We have added a few more explanations and references that hint how the current work can be connected to the application to the clean energy harvesting via nonlinear bulk photovoltaic effect [*Nature* **503**, 509–512 (2013); *Nat. Nanotechnol.* **5**, 143–147 (2010)] and scalable attosecond sources by heterostructuring multiple layers of 2D materials [*Nanophotonics* **12**, 255–261 (2023)]. We have also now cited the above mentioned critical works in the main text.

After incorporating suggested changes and improving the discussion, the present manuscript could be recommended for publication in Nature Communications.

We thank the reviewer again for the constructive and encouraging comments. We hope we have satisfactorily addressed the concerns of the reviewer.

Reviewer #3

The manuscript by J. Shi and H. Xu et al. reports strongly enhanced nonlinear optical responses in monolayer Janus-type topological semiconductors. The study combines a systematic comparison of four different transition metal dichalcogenides by three state-of-the-art nonlinear spectroscopy techniques: THz emission spectroscopy (TES) and second harmonic generation (SHG) probe second order nonlinear effects in opposite frequency regimes, whereas high harmonics generation (HHG) is employed to investigate higher order nonlinearities in the mid-infrared (MIR). Based on the MoS₂ family, the four investigated monolayers of 2D van der Waals materials are distinct by their presence or lack of topological band mixing (1T' vs. 2H structures) and degree of inversion symmetry breaking (top layer S → Se substitution for Janus-type compounds). The Janus 2D topological insulator (TI) 1T' MoSSe combines topological band properties with strong inversion symmetry breaking and therefore exhibits extremely enhanced TES and HHG, whereas its visible SHG is on the same order of magnitude as its topologically trivial 2H sibling. By additional variation of several experimental parameters and based on theory modelling, the authors thus find a drastic enhancement of the nonlinear susceptibilities for decreasing photon energies from 1 eV down to the THz frequency range; likely connected to the interband transition close to the new λ high symmetry points.

The research topic of monolayer 2D materials with THz and MIR optical nonlinearities is timely and highly relevant to contemporary material science and potential future application for detector or emitter devices. The manuscript is well written and concise. The vast amount of experimental and simulation data is presented in a clear and comprehensible way. Sample preparation and the variety of state-of-the-art methods are clearly not trivial and conducted in a reliable way. The main conclusions of the paper concerning a greatly enhanced nonlinear optic responses in the Janus-type topological compound 1T' MoSSe is plausible.

We thank the Referee for the succinct summary and for recognizing the novelty of our work and the advanced techniques developed for this study. We further appreciate the constructive critiques raised by the Referee.

Nevertheless, the manuscript could benefit from a clearer discussion on the influences of the four materials' distinct properties, e.g., by separating the role of the topological band texture and degree of inversion symmetry breaking.

We agree with reviewer that both topological band texture and a large degree of inversion symmetry breaking can make the NLO responses stronger. Specifically, 1T' phases generally have stronger NLO responses than 2H phases because of the topological band structure. Meanwhile, Janus structure have stronger NLO response than non-Janus structures because of the larger degree of inversion symmetry breaking. Hence, Janus 1T' MoSSe has the strongest NLO responses because it has both topological band texture and large degree of inversion symmetry breaking. This is confirmed in our experiments. To further clarify the influences of the topological properties on HHG efficiencies, we performed additional simulation using a two-band ($N_b = 2$) Su-Schrieffer-Heeger (SSH) model to demonstrate the topological enhancement in HHG, which is shown in Figure R1. The results confirm that the topological enhancement persists in higher order responses. To avoid repetition, we refer the Referee to pages 3 and 4 of this document for a detailed description of the new calculation. Briefly, the HHG is enhanced by 3 ~ 4 times when the SSH system is in the topological phase. And we have added this calculation results and details in the main text and Supplementary Note 4.

Moreover, the different techniques and the respective order of nonlinearities could be better explained in terms of their frequency realms: e.g., the $\chi^{(2)}$ of TES connecting NIR/VIS and THz wavelengths versus SHG depending on NIR/VIS and close-to-UV wavelengths. How do these frequency combinations relate to the bandgaps of the investigated materials?

We thank the Referee for the suggestions. The 1T' phase has bandgap at THz frequencies and the 2H-phase structure has bandgap in the visible range. We envision that irradiation with light frequency resonant with the bandgap could further enhance the nonlinear responses. We have added the bandgap information in the Table 1 to better visualize their relationship.

Overall, I believe that the work is suitable for a publication as an article in Nature Communications if the following minor remarks are addressed.

We thank the Referee for these positive comments. We have revised the manuscript based on the comments below.

A) Concerning the HHG experiments in Fig. 2: Why is the even order HHG stronger in 2H than in 1T' MoS₂, but opposite for Janus-type MoSSe? What do the asterisks on the HHG spectra indicate?

We thank the Referee for this helpful comment. First, 1T' MoS₂ does not have even order HHG due to the preservation of inversion symmetry. Hence, the even-order HHG in 2H phase MoS₂ turns out to be stronger than that in the 1T' phase. As for Janus structures, the inversion symmetry is broken in 1T' MoSSe, so its even-order HHG is higher than that of 2H MoSSe, because of the topological enhancement discussed above. The asterisks in Fig. 2b are artifacts due to high-order spectrometer diffraction. We have added this note to the main text.

B) Concerning the THz emission: The authors conclude primarily in-plane contributions from the THz emission polarization analysis in Fig. 3b. Is this conclusion justified when the x-polarization with in- and out-of plane contribution dominates the y-polarized component by about a factor of 3? Moreover, was the THz peak field as a function of the pump polarization (Fig. 3c) measured

for a specific emitted THz polarization (e.g. only for the Ex or Ey projections)? Or is it really the peak field independent of its polarization?

We thank the Referee for the comments. That is correct, it is based on the observation that the emission contains both x -polarized and a slightly weaker y -polarized components by a factor of 3. As we mentioned “Based on our experimental configuration, the x -direction emission has contribution from both in-plane and out-of-plane photoresponses while the y -direction emission originates only from the in-plane photoresponses.” The simultaneous observation of emission at both polarization proves the existence of the in-plane photoresponses, but we cannot rule out the out-of-plane contributions due to the experimental geometry used. For Fig. 3c, the THz peak field shown was measured without polarization projection. Thanks for the Referee’s careful reading and we have added a few lines to clarify the in-plane origin of the THz emission and the detection optics.

C) Comparing TES and SHG both relying on $\chi^{(2)}$ but with different frequency relations: Why is there significant SHG for 2H MoS₂ and MoSSe, but vanishing optical rectification leading to basically zero THz emission? A discussion on these subtle differences would be surely helpful.

We thank the Referee for this insightful comment. SHG concerns the sum-frequency process, which is characterized by $\chi^{(2)}(\omega, \omega; 2\omega)$, where ω is the input frequency. For 2H MoS₂ and MoSSe, the doubled frequency 2ω (around 3 eV) is above the bandgap (below 2 eV). Consequently, the SHG can be resonantly boosted. In contrast, the TES process is characterized by $\chi^{(2)}(\omega_1, \omega_2; \omega_1 - \omega_2)$, where ω_1 and ω_2 are two input frequencies, while $\omega_1 - \omega_2$ is the difference frequency and is in the THz range. In our experiments, ω_1, ω_2 and $\omega_1 - \omega_2$ are all below the bandgap of 2H MoS₂ and MoSSe. Hence, the TES for the 2H-phase structures is basically a non-resonant process and is intrinsically weaker than the resonant SHG. In addition, the signal levels in these two independent experiments are also influenced by the different detection sensitivities for TES and SHG. We have included these considerations in Supplementary Note 7.

D) According to Fig. 3f, the incident angle dependence seems to be dominated by the Janus structure with a minor influence of the topological properties. Is there a simple reason for that?

We thank the Referee for this critical comment. Yes, the angle dependence is dominated by the crystal structure. For example, if the crystal structure is isotropic with, e.g., octahedral symmetry, then there should be no angle dependence. On the other hand, if the symmetry of the crystal structure is lower, then the angle dependence can appear because electric fields with different polarization “feels” different atomic environment, and the optical responses would be different. On the other hand, topological properties typically do not have a significant impact on the incident angle dependence. For instance, isotropic structures, regardless of whether they possess topological properties or not, should not display any angle dependence. We further clarified this aspect in Supplementary Note 8.

E) According to the calculated second order nonlinear susceptibility of 2H MoSSe in Fig. 4e, the SHG should be nearly vanishing, which is in stark contrast to the experimental observation in Fig. 3e. Where does this contradiction come from? Is it related to different tensor elements not shown in Fig. 4e? Why did the authors choose to calculate these specific tensor elements shown in Fig. 4?

We thank the Referee for this helpful comment. The SHG of 2H MoSSe in Fig. 4e seems small because that of 1T' MoSSe is huge at small frequencies. In other words, the scale of the y-axis in Figure 4 is large. The nonlinear susceptibility $\chi^{(2)}$ of the SHG of 2H MoSSe is on the order of 1 nm/V, which is quite large (see also Figure R2). The SHG $\chi^{(2)}$ of typical crystals is on the order of 10^{-3} nm/V [see e.g., Shen YR (1984) *The Principles of Nonlinear Optics*. New York: Wiley].

The specific tensor elements shown in Fig. 4 are chosen because they are relatively large among all tensor elements. To further clarify this point, we calculate and plot the other in-plane elements of the SHG tensor in Figure R2. We have included the additional calculation and discussion in the main text, Supplementary Note 9 and Supplementary Fig. S15.

Figure R2. Non-zero in-plane components of the SHG tensor in (a) 1T' and (b) 2H MoSSe. Other in-plane components are forbidden by mirror symmetries. In (b), the magnitude of the SHG tensor is the same for $\chi_{yyy}^{(2)}$ and $\chi_{yxx}^{(2)}$ because of the three-fold rotation symmetry in the 2H phase.

F) The detector discussion in the end of the manuscript seems to come out of nowhere. In the main text, this should be better motivated in relation to the experimental results. Furthermore, the NEP abbreviation for noise-equivalent power is not defined in the main manuscript. For the broad audience of the targeted journal, the meaning of Fig. 4g should be also better explained.

The enhanced nonlinearity at THz and MIR frequencies highly indicates a practical approach for their sensing based on the nonlinear bulk photovoltaic effect, which has particular benefits of absence of dark current and high sensitivity. We agree with the Referee that more discussion would be needed to connect smoothly from the experiments to the detector application and the related Fig. 4g. We have added more discussion in the main text to clarify.

Reviewer #1 (Remarks to the Author):

The authors have addressed all my questions satisfactorily. However, in their revised version of the Supplemental Material, they now discuss the enhancement of the HHG response for SSH bulk. Unfortunately, there must be a mistake because it has been shown rigorously in Phys. Rev. A 102, 053112 (2020) that the HHG response is the same for the topological and trivial phase of SSH bulk as long as $(w^2 - v^2)^2$ is the same. Differences can only occur if the current was calculated in a way that is inconsistent with the Bloch ansatz used (or if edge states are explicitly taken into account, i.e., for a finite SSH chain and not for bulk). I suggest the authors remove that erroneous excursion on the SSH chain. Otherwise I recommend publication.

Reviewer #2 (Remarks to the Author):

The authors have looked at my suggestions carefully and addressed them satisfactorily. I am happy to recommend this manuscript for publication in Nature Communications.

Reviewer #3 (Remarks to the Author):

The revised manuscript by J. Shi and H. Xu et al. clearly improved and most of my previous remarks have been addressed. Therefore, I congratulate the authors for this comprehensive work and would like to recommend the manuscript for a publication as an article in Nature Communications. Before publication, nevertheless, the following two open remarks should be convincingly addressed, because they have not been fully resolved in the current state of the manuscript:

Concerning previous remark B:

I still do not understand the logic reasoning of concluding a "primarily in-plane origin of the TES signal" (in the authors' reply and on page 6 of the main manuscript). If the E_x component containing both in- and out-of plane origins is 3 times higher than the E_y component from a pure in-plane response, an out-of plane component could be even dominating! So in my eyes this conclusion is not sound at all and should be removed.

Concerning previous remark E:

The calculated second order response comparing 1T' and 2H MoSSe in Fig. 4e still seems to contradict the experimental results in Fig. 3e. The experimental ratio between 1T' and 2H does not even show a factor of 1.5, whereas Figs. 4e and R2 suggest at least 5 to 10 times increased second order response at 1.5 eV. How can this be reconciled? How reliable are the theoretical models if they do not even roughly match the experimentally observed ratios?

Reply to Reviewers

Reviewer #1:

The authors have addressed all my questions satisfactorily. However, in their revised version of the Supplemental Material, they now discuss the enhancement of the HHG response for SSH bulk. Unfortunately, there must be a mistake because it has been shown rigorously in Phys. Rev. A 102, 053112 (2020) that the HHG response is the same for the topological and trivial phase of SSH bulk as long as $(w^2 - v^2)^2$ is the same. Differences can only occur if the current was calculated in a way that is inconsistent with the Bloch ansatz used (or if edge states are explicitly taken into account, i.e., for a finite SSH chain and not for bulk). I suggest the authors remove that erroneous excursion on the SSH chain. Otherwise I recommend publication.

We thank the Referee for this insightful comment and have removed the discussions on the HHG of the SSH model. We believe there are some subtleties here in computing and comparing interband and intraband velocities for trivial vs. non-trivial phases but this is not a central aspect of the current paper and so we have followed the referees' suggestion.

Reviewer #2:

The authors have looked at my suggestions carefully and addressed them satisfactorily. I am happy to recommend this manuscript for publication in Nature Communications.

We thank the Referee for endorsing our publication in Nature Communications.

Reviewer #3:

The revised manuscript by J. Shi and H. Xu et al. clearly improved and most of my previous remarks have been addressed. Therefore, I congratulate the authors for this comprehensive work and would like to recommend the manuscript for a publication as an article in Nature Communications. Before publication, nevertheless, the following two open remarks should be

convincingly addressed, because they have not been fully resolved in the current state of the manuscript:

We sincerely appreciate the Referee for recognizing our work and supporting publication in Nature Communications.

Concerning previous remark B:

I still do not understand the logic reasoning of concluding a “primarily in-plane origin of the TES signal” (in the authors’ reply and on page 6 of the main manuscript). If the E_x component containing both in- and out-of plane origins is 3 times higher than the E_y component from a pure in-plane response, an out-of plane component could be even dominating! So in my eyes this conclusion is not sound at all and should be removed.

We thank the Referee for carefully reading our work and agree that the polarization analysis of the TES signal should be more clearly described. Based on our observation of the TES signal in the y -direction, which can only originate from in-plane photoresponses, we deduced that there are TES contributions from within the plane. However, when it comes to a stronger signal in the x -direction, which encompasses both in-plane and out-of-plane contributions, we cannot definitively determine each contribution as the referee correctly points out. To clarify, we have revised the sentence “Thus, the simultaneous observation of emission at both polarizations indicates a primary in-plane origin of TES signal but cannot rule out the out-of-plane contributions” to “Thus, the observation of emission in the y -direction indicates the existence of an in-plane current contributing to the TES signal but the stronger emission in the x -direction indicates there are likely significant contributions as well from out-of-plane currents. Further experiments are needed to disentangle the in-plane and out-of-plane contributions”. Per the Referee’s comments, we have also removed the sentence comparing with the theory since the out-of-plane contributions cannot be readily determined experimentally.

Concerning previous remark E:

The calculated second order response comparing 1T’ and 2H MoSSe in Fig. 4e still seems to contradict the experimental results in Fig. 3e. The experimental ratio between 1T’ and 2H does not even show a factor of 1.5, whereas Figs. 4e and R2 suggest at least 5 to 10 times increased second

order response at 1.5 eV. How can this be reconciled? How reliable are the theoretical models if they do not even roughly match the experimentally observed ratios?

We thank the Referee for pointing this out. We would like to remark that the theoretical calculations can only be considered qualitative estimations of the optical responses. Several issues can lead to inaccuracies in the calculations. The DFT calculations are intrinsically inaccurate regarding some electronic properties [see e.g., *Nat. Chem.* **8**, 831–836 (2016)], including bandgaps and interband/intraband velocities. This could lead to some errors in the optical response. In addition, the calculations of the optical responses performed in this work are based on the independent particle approximation (IPA). While IPA is widely adopted in optical response calculations [e.g., *Nature Commun.* **12**, 4330 (2021), *PNAS* **118**, e2100736118 (2021)], it (partially) ignores some many-body interactions, which can have an appreciable impact on optical responses [see, e.g., *Phys. Rev. B* **92**, 235432 (2015)].

From the experimental perspective, the detected SHG responses are also influenced by the sample quality, doping levels, etc. For example, in calculations, we assumed that the Fermi level falls within the bandgap (i.e., no doping), but in reality, there could be finite doping concentration, which would influence optical response. Although future developments with more advanced theoretical models and comprehensive experimental controls are desirable, the current qualitative theory-experiment comparison provides an informative testbed and guidance for such efforts. Thus, we added a few sentences in the main text to emphasize the future work needed to reach the quantitative comparison.

REVIEWERS' COMMENTS

Reviewer #1 (Remarks to the Author):

The authors followed my advice and removed the SSH part. I recommend publication.

Reviewer #3 (Remarks to the Author):

I appreciate the authors' convincing and honest replies. The recent changes of the manuscript sufficiently address the previously remaining critical points. Therefore, I am more than happy to recommend this exciting piece of experimental work for publication in Nature Communications. Congrats to all authors.

Our response: We would like to thank again all reviewers for their careful reading of our work and endorsing our publication in *Nature Communications*.

Reviewer #1 (Remarks to the Author):

The authors followed my advice and removed the SSH part. I recommend publication.

Reviewer #3 (Remarks to the Author):

I appreciate the authors' convincing and honest replies. The recent changes of the manuscript sufficiently address the previously remaining critical points. Therefore, I am more than happy to recommend this exciting piece of experimental work for publication in *Nature Communications*. Congrats to all authors.